# Geochemical Speciation, Uptake, and Transportation Mechanisms of Arsenic, Cadmium, and Lead in Soil–Rice Systems: Additional Aspects and Challenges

**DOI:** 10.3390/antiox14050607

**Published:** 2025-05-18

**Authors:** Chaw Su Lwin, Ha-il Jung, Myung-Sook Kim, Eun-Jin Lee, Tae-Gu Lee

**Affiliations:** Soil and Water Environment Division, National Institute of Agricultural Sciences, Rural Development Administration, Wanju 55365, Republic of Korea; chlwin1990@korea.kr (C.S.L.); msk74@korea.kr (M.-S.K.); eunjin0219@korea.kr (E.-J.L.); leetg7942@korea.kr (T.-G.L.)

**Keywords:** soil–rice systems, uptake, transporters, As, Cd, Pb

## Abstract

Potentially toxic elements (PTE), such as cadmium (Cd), lead (Pb), and arsenic (As), threaten rice (*Oryza sativa* L.) crop productivity and pose significant risks to human health when they are present in soil. This review summarizes the current understanding of soil and rice contamination with As, Cd, and Pb to provide an in-depth understanding of the dynamics of these contaminants and the mechanisms regulating their flow from soil to plants. It focuses on the following aspects: (1) these metals’ geochemical distribution and speciation in soil–rice systems; (2) factors influencing the transformation, bioavailability, and uptake of these metals in paddy soils; (3) metal uptake, transport, translocation, and accumulation mechanisms in rice grains; and (4) the roles of transporters involved in metal uptake, transport, and accumulation in rice plants. Moreover, this review contributes to a clearer understanding of the environmental risks associated with these toxic metals in soil–rice ecosystems. Furthermore, it highlights the challenges in simultaneously managing the risks of As, Cd, and Pb contamination in rice. The study findings may help inspire innovative methods, biotechnological applications, and sustainable management strategies to mitigate the accumulation of As, Cd, and Pb in rice grains while effectively addressing multi-metal contamination in paddy soils.

## 1. Introduction

The contamination of agricultural soils with potentially toxic elements (PTE) is a serious global concern [1,2,3,4]. PTE primarily enter agricultural soil ecosystems via atmospheric deposition from mining and industrial activities; the application of synthetic fertilizers (e.g., phosphate fertilizers), farmyard manure, and agrochemicals (e.g., pesticides); the introduction of sewage sludge and wastewater; and irrigation using contaminated groundwater [2,5,6]. Among PTE, arsenic (As), cadmium (Cd), and lead (Pb) are of particular concern because they frequently occur in the environment, which increases the likelihood for human exposure [7]. In fact, the Agency for Harmful Substances and Disease Registry and the U.S. Environmental Protection Agency list these three toxic elements among the top 10 PTE [8,9]. Their toxicity levels are dose-dependent, and chronic exposure to these PTE through food consumption has been linked to various health risks, including cancer [10,11]. They are non-biodegradable, persistent, and devoid of biological functions once they enter the environment.

Moreover, these PTE significantly degrade soil health and fertility by disrupting soil ecosystem functions, negatively affecting natural soil microbial communities, and altering soil chemical and physical properties [12,13]. Upon entering the edible parts of crops, these PTE interfere with various physiological and biochemical processes, including nutrient uptake, membrane stability, photosynthesis (through decreased chlorophyll production), ion homeostasis, and lipid peroxidation, which generate reactive oxygen species, ultimately leading to decreased plant biomass and grain yield [14,15,16]. Furthermore, edible crops with elevated concentrations of PTE pose significant risks to consumers and are unsuitable for consumption because of compromised food quality and safety.

Therefore, the mechanism behind the transfer of PTE from the soil to edible plant parts is a critical research area because it is directly related to food safety and security, with potential implications for human health. For instance, rice, which contains both dietary calories and toxic metals [17], is the primary staple food for approximately half of the global human population [18]. Rice-based agroecosystems play a critical role in global food security. The uptake and accumulation of toxic elements in rice grains is directly related to food safety and public health. Flooded paddy soils, in particular, create unique chemical conditions that can increase the mobility and bioavailability of certain PTE, especially As. According to the World Health Organization (WHO), Food and Agricultural Organization (FAO), and European Commission legislation, the maximum allowable limits for total As, Cd, and Pb are 300 µg kg^−1^, 200–400 µg kg^−1^, and 100–200 µg kg^−1^, respectively [19,20,21]. Therefore, understanding the behavior of PTE in rice-growing environments and their uptake pathways and accumulation in rice is essential for developing effective strategies to ensure the safety and sustainability of rice production.

This review focuses on three PTE—As, Cd, and Pb—owing to their high potential transfer from soil to the food chain. These toxic elements exhibit distinct biogeochemical behaviors, complicating the development of remediation strategies that can simultaneously decrease their toxicity and accumulation in agricultural products. Previous studies have introduced a simple and cost-effective method (chemical immobilization technology with soil amendments) for reducing metal bioavailability [4,22,23], with detailed explanations of its mechanisms for the remediation of contaminated arable soils. However, a single technique is often insufficient for simultaneously and effectively decreasing the bioavailability and uptake of As, Cd, and Pb in plants. Therefore, understanding the mechanisms of their uptake and translocation mechanisms in plants is essential for developing effective strategies to decrease their risks in soil–rice systems.

This review summarizes and discusses the geochemical distribution and speciation of As, Cd, and Pb in the soil, their uptake and transportation mechanisms from the soil to plant roots, and their subsequent movement into aerial plant parts. Furthermore, this review provides an in-depth understanding of the dynamics of As, Cd, and Pb in the soil; their controlling factors; their interactions with other metals, cations, and anions; and the mechanisms regulating their flow from soil to plants.

## 2. Literature Search Strategy

This review is based on a comprehensive search of the published literature using several electronic databases, including PubMed, Scopus, Web of Science, and Google Scholar. The search was conducted using a combination of specific keywords and corresponding terms found in titles and/or abstracts: “arsenic”, “cadmium”, “lead”, “rice”, “paddy soil”, “metal uptake”, “bioavailability”, “speciation”, and “metal transporters”.

Articles were included if they focused on rice systems and examined the behavior, transformation, bioavailability, or uptake of these toxic elements (As, Cd, and Pb) in paddy soils and rice plants. The initial screening process involved removing duplicate records, followed by a review of titles, abstracts, and full texts to exclude studies that were not relevant to the objectives of this review. No formal inclusion or exclusion criteria or systematic data synthesis methods were applied. Instead, the goal was to provide a broad overview and a critical discussion of key findings and emerging trends in the literature.

## 3. Geochemical Distribution and Speciation of As, Cd, and Pb in Paddy Soils

### 3.1. As

Arsenic (As) has an atomic weight of 74.9 and a specific gravity of 5.73. This semi-metallic element exists in the +5, +3, 0, and −3 oxidation states, depending on the prevailing physiochemical conditions of the environment. In soils, As can occur in the form of inorganic species, such as arsenate oxyanions [As(V)] and arsenite [As(III)], as well as methylated organic species, such as monomethylarsonic acid (MMA), dimethylarsinic acid (DMA), and trimethylarsinic acid (TMA) [24,25]. In contaminated soils, As exists primarily in two inorganic forms: As(V) and As(III). In addition, trace amounts of methylated organic As species, such as MMA and DMA, are also present in soils, often resulting from extensive pesticide, herbicide, and defoliant applications, as well as the microbial methylation of As [26,27]. Under aerobic conditions, As(V) is the predominant As species [28]. It can undergo hydrolysis and bind strongly to iron (oxy)hydroxides through inner-sphere bidentate and monodentate surface complexes [26]. However, under anaerobic conditions in flooded paddy soils, As(V) is reduced to As(III) through microbial activity. Moreover, As(III) is ten times more toxic and soluble than As(V) [29,30]. High concentrations of As(III) are dissolved in the soil solution under anaerobic conditions, leading to higher As concentrations in rice grains [29,31], as rice is typically cultivated under anaerobic conditions [32]. It was reported that As(V) accounts for approximately 73–96% of the total As content of aerobic soils [33], whereas As(III) accounts for approximately 87–94% of As in flooded soils. Although As(V) is less toxic than As(III), it is thermodynamically more stable under normal conditions and remains a major contaminant, owing to its toxicity and carcinogenicity to humans. The anthropogenic sources that contribute to elevated As concentrations in the soil include atmospheric deposition from metal mining; smelting; fossil fuel combustion; industrial dust and waste accumulation; the irrigation of wastewater and As-contaminated water originating from geogenic sources; and phosphate fertilizer, manure, pesticide, and herbicide application [34,35,36,37].

### 3.2. Cd

Cadmium (Cd) is a transition metal that has an atomic number of 48, an atomic weight of 112.4 g mol^−1^, and a density of 8.65 g cm^−3^. It predominantly exists as divalent Cd^2+^ ions and is mobilized under oxic and acidic conditions. Cd is more mobile and soluble than other toxic elements in the soil, making it readily available for plant uptake [38]. The behavior of Cd^2+^ in soil largely depends on the soil pH. At pH levels below 6.5, Cd^2+^ tends to form complexes with inorganic ligands (e.g., Cl^−^, SO_4_^2−^ etc.), while at pH levels above 6.5, it binds to iron oxides [39]. Additionally, a significant portion of Cd^2+^ binds to humic acids. In some soil types, CdCO_3_ is the dominant species; meanwhile, small quantities of Cd^2+^ sulfide species may also occur in other soil types. The primary sources of Cd^2+^ input into agricultural soils include the application of phosphate fertilizers, pesticides, wastewater irrigation, and biosolids, as well as the atmospheric deposition of dust and fumes from industrial activities, incineration, and mining [40,41,42,43,44].

### 3.3. Pb

Lead (Pb) has an atomic number of 82, an atomic weight of 207.19 g mol^−1^, and a specific gravity of 11.34 g cm^−3^. It primarily exists in the +2 oxidation state rather than in the +4 oxidation state. Pb^2+^ ranks second after As in terms of occurrence, human exposure potential, and toxicity. Inorganic Pb^2+^ is less mobile and soluble than As and Cd; therefore, rice absorbs minute concentration of Pb^2+^ [45,46]. Most of the absorbed Pb^2+^ ions are deposited in rice roots rather than being translocated to the aerial parts of rice. Pb^2+^ strongly binds to humic matter in organic soils and to iron oxides in mineral-rich soils. In flooded paddy soils, Pb^2+^ can bind to sulfur to form PbS precipitates; however, under oxidizing conditions, PbS releases Pb^2+^ ions. Pb^2+^ is released into agricultural soil ecosystems through various anthropogenic activities, including the use of leaded gasoline, fertilizers, and pesticides; the disposal of municipal solid waste; and the deposition of industrial dust and fumes [47].

## 4. Factors Influencing As, Cd, and Pb Distribution in Paddy Soils and Their Translocation in Rice

Not all PTE in the soil are available for uptake by rice; their availability depends on the type, concentration, and chemical forms of metals. Additionally, As, Cd, and Pb exhibit dynamic behaviors in paddy soils. Their bioavailability and uptake mechanisms are influenced by various factors, including soil physicochemical properties (e.g., soil pH and redox potential); plant root characteristics; As, Cd, and Pb forms and concentrations in the soil solution; environmental conditions; plant genotype; growth stages; soil amendment applications; and soil microbial dynamics [48,49,50,51]. Such a wide range of factors can significantly influence the bioavailability and uptake of toxic ions by rice, thereby affecting their accumulation in edible plant parts [52].

### 4.1. Soil pH

Soil pH significantly influences the concentrations of available As, Cd, and Pb in the soil, as well as their translocation to rice roots. As soil pH increases, the adsorption of cationic metals (Cd and Pb) tends to increase due to hydrolysis and the formation of MOH^+^ (M = Cd^2+^ or Pb^2+^), which, in turn, decreases their solubility and bioavailability. At high pH levels, several factors contribute to the decrease in bioavailable Cd and Pb in paddy soils: (1) an increase in negative charges on the soil surface, which enhances the adsorption of cationic metals (Cd and Pb); (2) the production of M(OH)^+^ precipitates due to hydrolysis; and (3) the formation of stable phases that bind to carbonate-bound, phosphate-bound, and Fe–Mn oxide-bound fractions [53]. It was also reported that increasing soil pH from 5.0 to 6.5 decreased the extractable phase of Cd by approximately 2.6 to 3 times [54]. In contrast, soil acidification increases the solubility of Cd, transforming it from a stable to a more bioavailable form [55]. Furthermore, Pb translocates to shoots when Pb ions cannot precipitate and are retained in plant cell walls due to low pH [56]. Conversely, oxyanionic As has the opposite solubility pattern. For example, the desorption of As increases with increasing soil pH, mainly because of proton consumption, which increases the number of anions on the exchange sites of the soil surface, along with both As(III) and As(V). It was reported that high soil pH (pH 8.5) increases negative surface charges [57], such as hydroxyl ions, thereby facilitating the desorption of As from Fe oxides. This, in turn, increases the mobilization of As at the root surface, leading to greater As accumulation in rice. Consequently, this condition enhances the desorption and mobilization of As at the root surface, ultimately increasing As accumulation in polished rice [58]. Therefore, the contrasting behaviors of oxyanionic As and cationic metals (Cd and Pb) in response to soil pH in paddy soils pose a significant challenge for simultaneously controlling the accumulation of these toxic elements in plants and rice.

### 4.2. Oxidation–Reduction Reactions (Redox Potential)

Redox potential is also a key driving factor that significantly and directly affects the solubility of As, Cd, and Pb in flooded paddy soils. Flooding and draining cycles during rice growing seasons cause substantial fluctuations in the redox potential, pH, and solubility of these toxic elements. The mobility and toxicity of As are highly dependent on the soil redox environment. The solubility of As increases dramatically as the redox potential decreases. For example, reduced redox potential facilitates the transformation of As(V) into As(III), which is more soluble and bioavailable than As(V) [59], because As(III) has a lower affinity for the soil solid phase. Conversely, As(V) readily adsorbs onto mineral components such as iron (hydr)oxides [60]. During drainage, As mobility decreases as As(V) becomes the predominant species [31].

By contrast, Cd and Pb are redox-insensitive toxic elements that do not change their oxidation states with respect to their solubility. However, the reductive dissolution of Fe and Mn hydr(oxide)s under decreasing redox potential in flooded soils can significantly increase the release and mobility of Cd and Pb through desorption reactions. Additionally, sulfide-rich conditions in reduced paddy soils promote the precipitation of Cd and Pb with sulfide minerals. For instance, the solubility of Cd decreases with lower redox potential due to the formation of CdS [61]. During drainage, increasing redox potential causes the dissolution of CdS [62] and the formation of water-soluble Cd sulfate (CdSO_4_), which, in turn, increases the solubility of Cd^2+^ in the soil solution, making it more available for rice roots to adsorb [63].

Sulfate reduction may also decrease the solubility of As, particularly in paddy soils that significantly produce Fe^2+^ under reducing conditions, which is likely due to the co-precipitation or sorption of As(III) by the newly formed FeS minerals [64]. Flooding conditions can also alter soil pH. Typically, soil pH in flooded paddy soils stabilizes within a neutral range. In flooded acidic soils, pH increases to a neutral level, owing to proton consumption during various reduction reactions. By contrast, alkaline soils tend to exhibit decreased soil pH due to CO_2_ accumulation during flooding [65]. The solubility of Cd and Pb decreases markedly as the redox potential decreases under flooded conditions because an increase in soil pH enhances the sorption of Cd^2+^ and Pb^2+^ ions.

Therefore, the solubility of As increases dramatically as the redox potential decreases, primarily because of the reduction of As(V) to As(III) and the attainment of neutral pH levels. When paddy soil is drained, the opposite processes occur rapidly, leading to increased Cd^2+^ and Pb^2+^ solubility and decreased As solubility [66,67,68].

### 4.3. Fe Plaque Formation and Radial O_2_ Loss

As rice is a semi-aquatic plant (moist or wet conditions), its roots have extensive longitudinal aerenchyma (a spongy tissue that forms spaces/air channels in plants) [69], which facilitates the transport of O_2_ from the shoots to the root tips, allowing it to thrive and extend its roots into flooded soils. Oxygen diffusion through the aerenchyma into the surrounding rhizosphere is referred to as radial O_2_ loss (ROL). ROL oxygenates the rhizosphere and promotes the oxidation of ferrous Fe (Fe^2+^) to ferric Fe (Fe^3+^). Consequently, yellow Fe (hydr) oxide precipitates, known as iron plaques, form on the root surface. These plaques primarily comprise amorphous or crystalline Fe (hydr)oxides [70]. As strong sorbents, Fe plaques hinder the uptake of As, Cd, and Pb by rice [70,71,72]. Several studies have reported that Fe plaques decrease metal concentrations in rice seedlings, with a negative correlation observed between As levels in rice grains and amorphous Fe oxide-bound As concentrations in the soil [73,74,75]. The role of Fe plaques in influencing As, Cd, and Pb concentrations varies depending on several factors, including the extent of plaque formation on the root surface, which differs among rice genotypes. In addition, Fe plaque formation is influenced by waterlogging conditions, O_2_ availability in the soil, the growth stages of rice plants, and the background concentration of Fe in paddy soils [76,77].

### 4.4. Rice Root Activity

As the first organ exposed to As, Cd, and Pb in the soil, rice roots serve as the first barrier against their toxicity. The root hairs and epidermal cells in the mature zone of the root tip absorb toxic metal ions from the soil. Root activity significantly influences the concentration and speciation of metal ions in the rhizosphere, thereby affecting the ability of roots to take up metal ions. To maintain the charge balance resulting from the absorption of nutrient ions, roots compensate by excreting H^+^ or OH^−^ ions [78]. For instance, metal ions are exchanged with H^+^ and OH^−^ ions that are produced during plant respiration and are adsorbed onto the surface of root epidermal cells [79]. Additionally, plant roots produce certain low-molecular-weight compounds and organic (e.g., dissolved organic C) or inorganic ligands (e.g., NO_3_^−^ and Cl^−^) that form soluble metal–ligand complexes [80]. Consequently, these complexes enhance the mobility of metal ions and facilitate their entry into the root epidermis as chelates. Other factors, such as root surface area, mycorrhizal relationships in the rhizosphere, transpiration pull, and the extent of the root system, can influence the solubility of toxic metal ions in the soil. Moreover, rice cultivars may affect the solubility of metals in the rhizosphere and their accumulation in grains. Different rice cultivars exhibit varying capacities for As, Cd, and Pb uptake, transport, and accumulation in the reproductive tissues [81].

### 4.5. Interaction with Cations (Ca^2+^, Mg^2+^, K^+^, Na^+^, and Mn^2+^) and Anions (NO_3_^−^, SO_4_^2−^, and PO_4_^2−^)

The transport pathways of Cd and Pb chemically resemble those of other divalent cations. Consequently, the presence of competing cations, such as Ca^2+^, Mg^2+^, or K^+^, in the soil medium can alter the amount of Cd^2+^ or Pb^2+^ concentrations taken up by rice roots because they share the same transporter proteins [82]. Na^+^ has been reported to enhance Cd uptake by plants [83,84]. Conversely, other metallic elements, such as Ca^2+^, Mg^2+^, and K^+^, inhibit Cd^2+^ and Pb^2+^ uptake by blocking their transport into rice roots [85]. Other divalent metal ions, such as Mn^2+^, Fe^2+^, Zn^2+^, and Si, also enter rice root cells through ion channels and carrier proteins and share the same pathways as Cd and Pb. These cations can inhibit Cd and Pb uptake through competitive interactions with these channels [41,86,87,88,89,90]. For instance, the excessive release of Mn owing to the dissolution of Mn oxides inhibits Cd absorption by rice roots. Mn oxides, which have strong oxidation and adsorption capacities, can influence the availability of Cd by interfering with its adsorption onto other metal oxides [41,91].

By contrast, anionic As primarily interacts with elemental S in paddy soils, especially under reducing conditions in which reduced S species bind to arsenite [As(III)]. In flooded soils, sulfate is readily reduced to dissolved sulfide (S^2−^) by sulfate-reducing bacteria. It was also reported that reduced S^2−^ immobilizes As through the formation of amorphous As–sulfide precipitates, such as AsS and As_2_S_3_ [92]. For instance, the high-sulfate treatment of rice plants has been reported to decrease the translocation of As from the roots to the shoots. Additionally, in the presence of Fe, S^2−^ can bind with Fe(II) to form FeS minerals, such as Fe_3_S_4_ and FeS_2_, thereby enhancing the adsorption of As(III) and reducing its mobilization [93]. In paddy soils, N predominantly exists in the form of ammonium (NH_4_^+^), with lower concentrations of nitrate (NO_3_^−^). The release of O_2_ from rice roots into the rhizosphere can promote the microbial nitrification of NH_4_^+^ near the root surface [94]. NO_3_^−^ concentrations can enhance the oxidation of As(III) and decrease its mobilization in paddy soils by increasing the abundance and activity of NO_3_^−^-dependent oxidizers [80,95]. The microbial oxidation of Fe(II) can occur simultaneously, forming insoluble Fe(III) precipitates that immobilize As(III) through co-precipitation and adsorption [96]. Furthermore, denitrification (NO_3_^−^ reduction) and Fe(II) oxidation can occur concurrently, facilitated by specific microbes that enhance As(III) adsorption, thereby decreasing As (III) availability under flooded conditions [97,98].

## 5. As, Cd, and Pb Uptake and Transport Mechanisms in Rice

Understanding the uptake and transport mechanisms of As, Cd, and Pb is essential for producing tolerant rice varieties with low metal accumulation through breeding or genetic engineering. Various transporter proteins are critical in facilitating the transport of free As, Cd, and Pb ions from soil solutions into rice. Transporters and intracellular high-affinity binding sites regulate metal uptake across cell membranes. Several protein families, including heavy metal P-type ATPases (HMA), natural resistance-associated macrophage proteins (NRAMP), cation diffusion facilitators (CDF), Zn–Fe permease proteins (ZIP), and ATP-binding cassettes (ABC), have been implicated in metal ion transport in plants [99]. The transport of As, Cd, and Pb involves five major processes: (i) uptake by the roots; (ii) translocation to the xylem; (iii) transport to the shoot via the xylem; (iv) redistribution through intervascular transfer at the node; and (v) transport to rice grains from leaf blades via the phloem. Additionally, rice plants can absorb metal ions through their aerial parts, such as leaves. These aforementioned processes determine the accumulation of As, Cd, and Pb in rice grains [100].

Rice roots act as the primary entry point for metal ion uptake and serve as the first barrier to the absorption of toxic elements [101,102]. Metal ions adsorbed onto the surface of the root epidermis enter root cells via rhizosphere interactions, root membrane transporters, and ion channels. Once inside the root, metal ions move from the epidermis to the cortex, stele, and xylem for upward transport to the shoots. This movement occurs via two primary pathways: the apoplastic pathway (via extracellular spaces, including the cell wall, interstitial spaces, and conduit cavities) and the symplastic pathway (cell-to-cell movement mediated by polarized influx and efflux transporters) [103,104]. In the apoplastic pathway, Casparian strips act as barriers in the endodermis, limiting metal ion movement to the xylem. Meanwhile, the symplastic pathway is slower and involves intracellular flow through the plasmodesmata. Metal ions cross the endodermis, enter the stele, move into the parenchyma cells through the pericycle sheath, and finally enter the xylem [105]. During this process, a fraction of metal ions is sequestered into root cell vacuoles via specific transporter proteins before entering the xylem. Once in the xylem, metal ions are transported to aboveground plant parts, including leaves, stems, and nodes, driven by transpiration pull and root pressure. The efficiency of metal ion translocation depends on the sequestration capacity of the vacuole and efficiency of xylem loading. High levels of metal ion accumulation in rice grains are primarily associated with phloem transport, as the phloem redistributes metal ions within the aboveground parts of the plant [106,107]. Thus, metal ions absorbed by the roots are first translocated to the aboveground parts through the xylem and are subsequently transported to rice grains via the phloem [108,109]. The journey of As, Cd, and Pb from soil to rice grain is presented in Figure 1.

### 5.1. As Transport in Soil–Rice Systems

Under flooded conditions, As(III) is the most dominant As species in the soil, and rice roots rapidly take up As(III), primarily through the influx transporter OsNIP2;1 [50,101,110]. OsNIP2;1, also known as Si influx transporter (Lsi1), primarily functions as a Si transporter and serves as the major pathway for As(III) entry into rice roots. Lsi1 is strongly expressed in rice roots, and mutations in this transporter significantly decrease As(III) influx into rice roots [110]. Lsi1 is localized in the plasma membrane on the distal side of both endodermal and exodermal cells, where Casparian strips occur. Although other NIP family members, such as OsNIP1;1, OsNIP2;2, and OsNIP3;1, are expressed at very low levels, they are unlikely to play a significant role in As(III) influx [50].

After entering the root cells, a portion of As(III) is detoxified via sequestering into vacuoles following complexation with thiol compounds, such as phytochelatins (PCs). This process occurs predominantly in the pericycle and endodermal cells of rice roots [71]. The transport of As(III)–PC complexes into vacuoles for further detoxification is catalyzed by *Oryza sativa* ABC transporter (OsABCC1) [111], which are crucial in limiting As translocation by sequestering As(III)–PC complexes into vacuoles, wherein As is stored and distributed in plants. As OsABCC1 is expressed in various plant parts, including the roots, leaves, and nodes, removing it will significantly increase As accumulation in rice grains [112].

While Lsi1 is responsible for As(III) uptake into rice roots, the Si efflux transporter (Lsi2) mediates As(III) translocation to the xylem [110,113]. Lsi2 is localized on the plasma membrane of exodermal and endodermal cells in the roots; however, unlike Lsi1, it is situated on the proximal side [114]. The pathway of As from the soil medium to the stele involves uptake through the aquaporin channel Lsi1 and efflux toward the stele mediated by Lsi2 [113,114]. Lsi2 mutation significantly decreases As accumulation in rice shoots by 66–75% [110]. Similarly, knocking out Lsi1 decreases As concentration in the straw, although it does not significantly affect the grains. By contrast, knocking out Lsi2 leads to a significant decrease in As accumulation in both the straw and grains compared to wild-type rice. These findings suggest that Lsi2 is more critical than Lsi1 in regulating As transport and accumulation in rice shoots and grains. However, Lsi2 disruption also impairs Si uptake, which negatively affects rice growth and can decrease grain yield by up to 60% [113].

When paddy soils are flooded, certain concentrations of As(V) are present due to the oxidation of As(III) coupled with the denitrification and release of oxygen by plant roots [99]. In this case, the As(V) species is taken up by rice roots through phosphate transporters, such as OsPT1, OsPT4, and OsTP8. Once inside the root cells, As(V) is rapidly reduced to As(III) via the activity of transporters such as OsHAC1;1, OsHAC1;2, and OsHAC4 [110]. The resulting As(III) is transported to the xylem sap via Lsi2. Consequently, As(III) is the primary form of As in plant cells [115,116]. Notably, the overexpression of OsHAC1;1, OsHAC1;2, and OsHAC4 significantly increases the efflux of As(III) from the root cells, thereby reducing As accumulation in rice.

Aside from inorganic As species, several methylated As species, including methylated As species such as MMA and DMA, are present in the soil. While rice plants lack the As methylation ability, methylated As species in rice are derived from the soil by microbial activity after soil flooding and additions of organic matter [117]. Although rice roots can absorb methylated As species mediated by certain transporters like Lsi1 [118], their uptake by rice plants is slower than that of As(III). However, organic As species are much easier and 10 times more efficiently translocated to the shoots and reproductive parts than inorganic As species via the xylem and phloem [119,120]. Additionally, DMA does not form complexes with PCs or become sequestered in vacuoles. Consequently, DMA remains highly mobile within plants [121]. The putative peptide transporter (OsPTR7) in rice has been identified as being involved in DMA accumulation in rice grains [122].

### 5.2. Cd Transport in Soil–Rice Systems

Cd is highly mobile in the soil solution, and the root uptake of Cd begins when Cd ions reach the root surface [123]. Rice roots primarily take up Cd ions in the forms of Cd^2+^ and CdCl^−^. The uptake of Cd^2+^ is mainly facilitated by the Mn transporter *Oryza sativa* NRAMP5 (OsNRAMP5) [79]. The NRAMP family, known as the transporters of divalent transition metals, is present across all living organisms [124]. OsNRAMP5 is predominantly expressed in rice roots and is polarly localized to the plasma membranes on the distal sides of both exodermal and endodermal cells [79]. The expression of OsNRAMP5 promotes Cd^2+^ uptake, leading to increased Cd accumulation in rice grains [125]. Hence, knocking out OsNRAMP5 significantly decreases Cd^2+^ uptake by rice roots and decreases Cd accumulation in shoots and grains [126]. Other transporters, such as OsNRAMP1, *Oryza sativa* iron-regulated transporters (OsIRT1 and OsIRT2), and OsCd1 (a member of a major facilitator family), are also involved in Cd uptake; however, their contributions are minor compared to those of OsNRAMP5 [127,128]. Studies have shown that OsIRT1 and OsIRT2 influence the root absorption of Cd [129], whereas OsCd1 also contributes to the uptake of Cd by rice roots [130].

After being transported into rice roots, a fraction of the Cd is sequestered into root vacuoles via the *Oryza sativa* heavy metal ATPase family protein (OsHMA3) [131]. Variations in the coding sequence of OsHMA3 across rice cultivars affect Cd sequestration and transportation, resulting in differences in Cd accumulation among cultivars [132]. OsHMA3, which is predominantly expressed in root cells, sequesters Cd in vacuoles and regulates its loading into the xylem [131]. Additionally, the ABC family member OsABCC9 contributes to Cd sequestration in root vacuoles [43,133]. As OsABCC9 is localized in the tonoplast of parenchyma cells, knocking it out results in increased Cd sensitivity in rice plants and elevated Cd levels in the xylem sap and grains [130,134].

Cd translocation from roots to shoots is facilitated by transporters such as OsHMA2 and OsZIP7. OsHMA2, which is localized in the plasma membrane of the root pericycle and parenchyma cells of vascular bundles in the nodes, is essential for delivering Cd from roots to developing tissues [135,136,137]. Hence, knocking out these genes decreases the inward flow of Cd [138]. Additionally, OsHMA2 is localized in the phloem of the vascular bundle and mediates Cd transport through the phloem into growing structures. Knocking out OsHMA2 significantly decreases Cd translocation to rice grains; moreover, it also disrupts Zn transport and affects plant growth [139]. Similarly, OsZIP7 is involved in xylem loading and transporting Cd upward to the phloem [137].

Node-expressed transporters such as OsCCX2 also mediate Cd translocation. OsCCX2, a plasma membrane-localized Ca^2+^ transporter, regulates Cd transport between roots and stems. Thus, knocking out this gene decreases Cd concentrations in rice grains [140]. Additionally, OsCAL1, a defensin-like protein expressed mainly in the root exodermis and xylem parenchyma cells, chelates Cd for secretion into the extracellular spaces, thereby reducing cytosolic Cd levels and facilitating root-to-shoot transport via xylem vessels [141]. Cd transport into rice grains occurs primarily via phloem transport, which is regulated by low-affinity cation transporters (OsHMA2 and OsLCT1). OsHMA2 is responsible for reloading Cd from parenchymal cells into the phloem of diffuse vascular bundles [125]. OsLCT1, a phloem plasma membrane protein localized in the uppermost node of the vascular bundle, is involved in Cd translocation through the phloem [142,143]. OsLCT1 expression increases significantly during grain maturity, which is a critical period for Cd accumulation in rice grains [142].

### 5.3. Pb Transport in Soil–Rice Systems

Rice roots serve as the initial point for facilitating the entry of Pb into the plant. Adsorbed Pb passively penetrates the roots and follows the water transport channels [51]. Once Pb is absorbed and passively accumulated in the roots, it utilizes the apoplastic pathway and is subsequently translocated through the vascular flow system. However, the specific transport mechanisms, including the roles of the transporter proteins responsible for stimulating Pb uptake in rice, remain unclear. Some studies have pointed out that plants can take up Pb via Ca^2+^-permeable channels [144]. Additionally, similar to divalent Cd^2+^, Pb^2+^ uptake is linked to the NRAMPs family of transporters in rice [90]. OsNRAMP5 has been identified as a major transporter for Pb^2+^ uptake, similar to its role in Mn^2+^ and Cd^2+^ uptake. Short-term hydroponic studies have shown that knocking out OsNRAMP5 Pb accumulation in rice grains by 50% significantly decreases its accumulation in roots and shoots. Moreover, studies have confirmed that the expression of transporter genes, including OsNRAMP5 and OsHMA6, is induced by Pb exposure in rice [145].

HMA facilitates metal detoxification. For example, OsHMA9 is predominantly expressed in vascular tissues (phloem and xylem) and contributes to Pb uptake and transport within and from these tissues [90]. Additionally, OsHMA2, OsHMA3, and OsHMA4 are implicated in Pb loading. These genes exhibit significantly higher expression in rice roots than in shoots and leaves, which likely explains increased Pb accumulation in the roots and decreased translocation to the shoots [146]. Furthermore, the ABCC1 transporter gene is expressed in rice, particularly in the roots of Pb-stressed plants, where it facilitates PC–Pb complex formation and vacuolar sequestration [146].

As Pb^2+^ is a divalent metal cation, it may also enter plant cells through channels or transporters associated with other divalent transition metals such as Cd, Co, Fe, Ni, and Zn. However, further studies are required to elucidate the precise mechanisms underlying Pb uptake in rice. Pb^2+^ is less mobile in the soil–rice system than Cd and As. Most Pb is retained in the root tissues, with only a small portion translocated to the aerial parts through transporters [147]. Several factors limit Pb transport from the roots to the aerial plant parts, including immobilization by negatively charged pectins in the cell wall [48,148,149], the precipitation of insoluble Pb salts in intercellular spaces [150], accumulation in the plasma membrane [151], and sequestration in the vacuoles of rhizodermal and cortical cells.

## 6. Key Challenges and Future Directions

Several studies have been conducted on the uptake, toxicity, and oxidative stress in rice exposed to high As, Cd, and Pb concentrations. Additionally, other researchers have attempted to develop various mitigation strategies, such as incorporating soil amendments to decrease metal bioavailability and limit plant uptake by modifying the soil environment, releasing essential plant nutrients, and developing rice varieties that can tolerate or avoid the accumulation of HM ions by enhancing or modifying transporter proteins. Numerous studies have demonstrated that agricultural soils are exposed to multiple PTE. Thus, the toxicological consequences of exposure to a mixture of metals differ significantly from those of exposure to individual metals. Therefore, the process and need for specific adaptations that simultaneously minimize metal uptake and translocation to the edible parts of rice are complicated. Information and research on developing new rice varieties that can tolerate multi-metal contamination is still lacking, which is one of the future challenges in mitigation strategies and research.

Metal(loid)s behave differently in the soil, affecting their availability, uptake, and transport to plants. In plants, the absorption of metal ions is facilitated by their respective transporters, and once inside the cells, different organelles utilize distinct transport proteins according to their metal needs and biological functions. Due to the chemical similarities between PTE such as As, Cd, and Pb and essential minerals, several transporters that manage nutrient uptake may also inadvertently allow for heavy metal uptake. Thus, creating or modifying transporter proteins that selectively exclude As, Cd, and Pb without interfering with the uptake of essential minerals is difficult.

Additionally, the bioavailability of As (metalloid), Cd, and Pb (metals) in flooded paddy soils varies widely because of factors such as pH, redox potential, organic matter content, and microbial interactions. A rice variety that is tolerant to metals in one type of soil may not perform well under different soil conditions. Therefore, specific soil and water management practices must be selected under metal-enriched conditions.

Effectively addressing multi-metal contamination in soils and preventing harmful translocation in plants may require a combination of genetic engineering, microbial support, and soil amendment strategies. Balancing HM tolerance with crop productivity and nutritional quality is a long-term goal, particularly under diverse environmental conditions. Developing new rice varieties that can tolerate multi-metal contamination presents significant challenges owing to the complexity of metal uptake and translocation pathways, environmental variability, and the need to maintain crop yield and quality. One major hurdle is designing transporter proteins that can exclude toxic metals, such As, Cd, and Pb, while allowing essential nutrients to enter the plant. Additionally, achieving tissue-specific exclusion, in which metals are sequestered in roots or shoots rather than in grains, requires advanced gene-editing techniques and detailed knowledge of regulatory networks. Multi-metal tolerance also demands balance; tolerance to one metal may increase sensitivity to others or decrease nutrient uptake, making it difficult to engineer effective broad-spectrum resistance. Furthermore, rigorous field trials across diverse environmental conditions are needed to validate laboratory results, whereas statutory approval and public acceptance of genetically modified or gene-edited rice can delay development. These complexities underscore the need for continued research, collaboration, and innovation in breeding and biotechnological approaches to develop safe, high-yield, and resilient rice varieties for growth in contaminated soils.

## 7. Conclusions

The contamination of paddy soils with As, Cd, and Pb, as well as the uptake and translocation of these PTE to rice, have raised significant environmental and food safety concerns. This review highlights the complex interactions among soil properties, metal speciation, and uptake and translocation mechanisms that govern the mobility and accumulation of As, Cd, and Pb in rice. While considerable progress has been made in understanding the uptake, translocation, and detoxification of these potentially toxic elements (PTE), managing multi-metal contamination remains a significant challenge. Future strategies should integrate soil and water management, the genetic improvement of rice varieties with selective metal exclusion or sequestration traits, and effective regulations to decrease the sources of these toxic elements to the rice ecosystems. Advancements in biotechnology and breeding approaches can offer promising tools for developing resilient and high-yielding rice varieties suited to contaminated environments, but sustained interdisciplinary efforts will be critical to ensure food safety and security under growing soil environmental pressure.

## Figures and Tables

**Figure 1 antioxidants-14-00607-f001:**
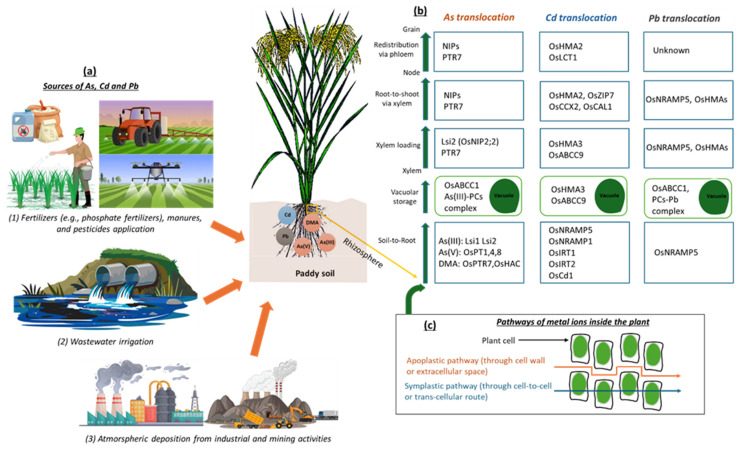
Summary of the distribution and direction of As, Cd, and Pb transportation in soil–rice systems. (**a**) Possible anthropogenic sources of As, Cd, and Pb in paddy soil. (**b**) Summary of the contribution of transporter genes in As, Cd, and Pb translocation from the roots to the aboveground parts of rice and, finally, accumulation in the grain. (**c**) Pathways of As, Cd, and Pb transportation in rice plant first occurs through root absorption via apoplastic and symplastic routes. Once metal ions enter the root cells, they are sequestered in root vacuoles or transferred to the xylem and then transported upward via xylem loading, transferred from the root to the shoot via the xylem, and redistributed to the grain via the phloem after reaching the node.

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
