# Peer review of "Geochemical Speciation, Uptake, and Transportation Mechanisms of Arsenic, Cadmium, and Lead in Soil–Rice Systems: Additional Aspects and Challenges"

_antioxidants, 2025, doi:10.3390/antiox14050607_

Round 1

Reviewer 1 Report

A useful overview of the current status of our knowledge of the mobilization of three potentially toxic elements from soil into rice grain.  A clear explanation of the challenges associated with the simultaneous decrease in the contamination of rice grains.

Overall this is a well-constructed review of the topics outlined by the title.  I have a number of detailed comments.

The term “heavy metal” is strongly discouraged.  IUPAC set out their position over 20 years ago.  Here is a more recent article whose recommendation (use “potentially toxic element, PTE,” instead) should be adopted in this manuscript. Int J Environ Res Public Health. 2019 Nov 13; 16(22): 4446. doi: 10.3390/ijerph16224446.

Also to be discouraged is the use of the element symbol as an abbreviation for the name of the element.  Especially when arsenic is involved as readers may have to back track to distinguish between As = arsenic and As = because.  But in a scientific article you may want to distinguish between the elemental form and some species, when it would be logical to use the symbol As to mean elemental arsenic and not just as a shorthand for all species containing arsenic.  For example in the sentence “this review provides an in-depth understanding of the dynamics of As, Cd, and Pb in the soil,” readers might well think you are talking about the elements, where as you really mean “the relevant compounds of arsenic, cadmium and lead.”   Also examine your use of “reduce” and “reduction” and replace with “decrease” for those situations when you are not talking about a change in oxidation number (i.e. chemical reduction).  And examine your use of “amount.”  There are several instances in the manuscript where you really mean “concentration.”

Introduction: it is not only humans that are risk from the consumption of contaminated grain, but also livestock that are fed rice straw. You should clarify that your review is really about grains for human consumption.  Nowhere in your review do you give any indication of the concentrations of the relevant elements and species are in rice.  Some numbers indicative of the range of total element concentrations might be useful for readers and maybe also an explanation that (unlike the situation for seafood) all of the arsenic compounds that are consumed after cooking are potentially harmful, though not all are known human carcinogens.

Line 57.  “Risk” is not the right word.  “Potential” maybe.

Lines 60 – 63.  Your summary suggests that the problem has been solved.  What is the evidence for stating that “However, a single technique is often insufficient to effectively reduce [i.e. decrease] the bioavailability and uptake of As, Cd, and Pb in plants.  Did all three of these papers describe the same method?  If not then “method” should be plural.

Line 80.  There are lots of soils in the US that are contaminated with a legacy of arsenic-containing agrochemicals (one of the reasons why some US rice has a relatively high concentration of DMA).  What about soils in Vietnam that were sprayed with Agent Blue?  Move the later material from around line 95 to here.

Line 85.  Delete “tends.”

Line 88.  Need a citation to support this statement about relative toxicity.

Also what about irrigation with ground water contaminated  from naturally occurring arsenic ( as in many parts of West Bengal, India and Bangladesh)?

Line 104.  Give examples of these ligands.

Line 157.  Would this not apply to all plants?

Line 192.  The first example of a rather technical term.  To what extent will your readers understand all the botanical terminology?  Maybe you should you include a glossary of terms?

Line 198.  An example of the potential for confusion by using As to stand for “arsenic.”

Line 217.  Does this apply to arsenic?

Line 239.  The first use of a researcher’s name.  Seems odd. Any particular reason? You only do this four times out of 133 papers cited.

Line 255.  I don’t think “Understanding” can serve as a “biological marker”. A biological marker is a measurable characteristic that can indicate normal or abnormal biological processes, or a response to an exposure or intervention.

Lines 323 324.  I don’t think these compounds are genes, they are enzymes that catalyze the reduction.

Line 330.  I don’t think MMA and DMA are volatile.

Line 337.  You should discuss whether rice can methylate arsenic species (and provide relevant citations) and this might be a good place to do this.

Line 341.  CdCl should carry one positive charge.

Line 344.  Is “polarly” the right word?

Line 384.  This sentence does not make sense.  Maybe replace “soil” with “plant”?

Lines 389, 390.  This sentence doesn’t make sense and needs revision.  “Some studies have suggested that Pb uptake occurs partly via Ca2+-permeable channels, similar to K+ and Ca channels.”  .

Lines 416, 417.  This sentence is problematical in that it is too vague.

Line 430.  This is ambiguous.  Not interact with each other, but interact with the components of the soil—right?

Line 432.  “Correspond” is not the right word.

Line 437.  Delete “and must be challenging”.

Lines 453 and 457.  The word “regulatory” is used with two quite different meanings.  The second one should be replaced with “statutory” or “government.”

The conclusion repeats statements that have already been made and could be more concise.

Author Response

Here is a draft of the point-by-point response to the reviewer's comments:

1. The term “heavy metal” is strongly discouraged.  IUPAC set out their position over 20 years ago.  Here is a more recent article whose recommendation (use “potentially toxic element, PTE,” instead) should be adopted in this manuscript.Int J Environ Res Public Health. 2019 Nov 13; 16(22): 4446. doi: 10.3390/ijerph16224446.

Answer: Thanks for your useful comments. As recommended, we will replace “heavy metal” with “potentially toxic element (PTE)” throughout the manuscript.

2. Also to be discouraged is the use of the element symbol as an abbreviation for the name of the element.  Especially when arsenic is involved as readers may have to back track to distinguish between As = arsenic and As = because.  But in a scientific article you may want to distinguish between the elemental form and some species, when it would be logical to use the symbol As to mean elemental arsenic and not just as a shorthand for all species containing arsenic.  For example in the sentence “this review provides an in-depth understanding of the dynamics of As, Cd, and Pb in the soil,” readers might well think you are talking about the elements, where as you really mean “the relevant compounds of arsenic, cadmium and lead.” 

Answer: We acknowledge the potential confusion caused by using element symbols instead of names, especially for arsenic (As). We will revise the manuscript to ensure clarity and consistency in naming the elements.

3. Also examine your use of “reduce” and “reduction” and replace with “decrease” for those situations when you are not talking about a change in oxidation number (i.e. chemical reduction).  And examine your use of “amount.”  There are several instances in the manuscript where you really mean “concentration.”

Answer: We will carefully examine our use of "reduce" and "reduction" and replace them with "decrease" when referring to changes unrelated to oxidation-reduction reactions. In cases where "amount" was used inappropriately, we will replace it with "concentration" for more precise scientific communication.

4. Introduction: it is not only humans that are risk from the consumption of contaminated grain, but also livestock that are fed rice straw. You should clarify that your review is really about grains for human consumption.  Nowhere in your review do you give any indication of the concentrations of the relevant elements and species are in rice.  Some numbers indicative of the range of total element concentrations might be useful for readers and maybe also an explanation that (unlike the situation for seafood) all of the arsenic compounds that are consumed after cooking are potentially harmful, though not all are known human carcinogens.

Answer: To explicitly clarify that our study focuses on rice grains for human consumption, we will revise the introduction to remove any ambiguity regarding livestock feed. We have revised and added more details in the Introduction section.

5. Line 57.  “Risk” is not the right word.  “Potential” maybe.

Answer: We have revised it.

6. Lines 60 –63.  Your summary suggests that the problem has been solved.  What is the evidence for stating that “However, a single technique is often insufficient to effectively reduce [i.e. decrease] the bioavailability and uptake of As, Cd, and Pb in plants.  Did all three of these papers describe the same method?  If not then “method” should be plural.

Answer: In our previous articles, we discussed chemical immobilization technology using various immobilizing agents, as it is one of the most promising remediation techniques for contaminated arable soils. Throughout all our papers, we consistently referred to the same approach—chemical immobilization.

7. Line 80.  There are lots of soils in the US that are contaminated with a legacy of arsenic-containing agrochemicals (one of the reasons why some US rice has a relatively high concentration of DMA).  What about soils in Vietnam that were sprayed with Agent Blue?  Move the later material from around line 95 to here.

Answer: We have revised these sentences properly. 

8. Line 85.  Delete “tends.”

Answer: We have revised it. 

9. Line 88.  Need a citation to support this statement about relative toxicity.

Answer: We have added relevant citation for this statement. 

10. Also what about irrigation with ground water contaminated  from naturally occurring arsenic ( as in many parts of West Bengal, India and Bangladesh)?

Answer: We have added the information. 

11. Line 104.  Give examples of these ligands.

Answer: We have added the examples of ligands in the text.

12. Line 157.  Would this not apply to all plants?

Answer: We have revised this sentence properly. Yes, this can apply for all plants.

13. Line 192.  The first example of a rather technical term.  To what extent will your readers understand all the botanical terminology?  Maybe you should you include a glossary of terms?

Answer: We have revised them properly. 

14. Line 198.  An example of the potential for confusion by using As to stand for “arsenic.”

Answer: We have revised it properly. 

15. Line 217.  Does this apply to arsenic?

Answer: We have properly revised this sentence. Yes, this also applies to arsenic. For example, dissolved organic carbon can compete with arsenic for sorption sites in the soil, leading to increased arsenic availability.

16. Line 239.  The first use of a researcher’s name.  Seems odd. Any particular reason? You only do this four times out of 133 papers cited.

Answer: We have properly revised it. 

17. Line 255.  I don’t think “Understanding” can serve as a “biological marker”. A biological marker is a measurable characteristic that can indicate normal or abnormal biological processes, or a response to an exposure or intervention.

Answer: We have properly revised this sentence. 

18. Lines 323 324.  I don’t think these compounds are genes, they are enzymes that catalyze the reduction.

Answer: We have properly revised it. There are transporter proteins encoded by genes.

19. Line 330.  I don’t think MMA and DMA are volatile.

Answer: We have deleted this sentence and added more relevant information. Yes, they are not volatile but can be reduced to volatile arsine forms. 

20. Line 337.  You should discuss whether rice can methylate arsenic species (and provide relevant citations) and this might be a good place to do this.

Answer: We have added relevant information with citation. 

21. Line 341.  CdCl should carry one positive charge.

Answer: We have checked this compound carefully. 

22. Line 344.  Is “polarly” the right word?

Answer: We have revised it properly. Yes, it localized in a polar manner to the plasma membrane. 

23. Line 384.  This sentence does not make sense.  Maybe replace “soil” with “plant”?

Answer: We have revised it. 

24. Lines 389, 390.  This sentence doesn’t make sense and needs revision.  “Some studies have suggested that Pb uptake occurs partly via Ca2+-permeable channels, similar to K+ and Ca channels.”.

Answer: We have revised this sentence properly. 

25. Lines 416, 417.  This sentence is problematical in that it is too vague.

Answer: We have deleted this sentence. 

26. Line 430.  This is ambiguous.  Not interact with each other, but interact with the components of the soil—right?

Answer: We have revised this sentence properly. 

27. Line 432.  “Correspond” is not the right word.

Answer: We have corrected it properly. 

28. Line 437.  Delete “and must be challenging”.

Answer: We have deleted it. 

29. Lines 453 and 457.  The word “regulatory” is used with two quite different meanings.  The second one should be replaced with “statutory” or “government.”.

Answer: We have replaced it. 

30. The conclusion repeats statements that have already been made and could be more concise.

Answer: The conclusion will be revised to not only emphasize the development of rice varieties that can limit or sequester these toxic elements (As, Cd, and Pb) but also highlight the critical importance of reducing the input of these contaminants into rice agrosystems.

We thank the reviewer again for the valuable feedback and stand ready to address any further comments or suggestions.

Reviewer 2 Report

No systematic approach to do the review, for instance PRISMA, was used. No methodology for the review is described in the paper. This is a major issue.

This review deals with a very important topic when it comes to rice fields, humid agrosystems that, due to their characteristics, can promote the passage of trace elements (toxic elements) to the crop and, consequently, affect the population. So, initially, this is so positive.

However, I would like to make some comments trying to improve the paper for its publication, in the case it will be accepted.

My first suggestion for the introduction is that you can pointed the relevance of the rice agrosystems and the production of rice to feed people, as it is one of the main cereals and basic for many countries. Additionally, it is important to remember that As is an essential element (micronutrient) for humans, the toxicity depends on the dose. (https://www.science.org/content/blog-post/arsenic-rather-lot-when-you-think-about-it) (https://pubmed.ncbi.nlm.nih.gov/18174893/).

In the second section, there are several mistakes that should be corrected or clarified.

For instance (line 75): “This semi-metallic element exists in the +3, 0, and +5 oxidation states,” this is wrong, the oxidation states are +5, +3, 0 and -3. See IUPAC table of oxidation states (https://i0.wp.com/www.compoundchem.com/wp-content/uploads/2015/11/The-Periodic-Table-Of-Oxidation-States-2016.png?ssl=1).

Line 100: Please correct ”Cd2+ has an atomic number of 48, an atomic weight...” It maybe Cd has an...., not the cation.

Line 112:, the same comment applied to Pb: Pb has an atomic....

Section 3 is the most important in the work. In general, the content is so general and probably can be applied to other metals and metalloids. However, the references centered in these three elements are enough good.

Line 207: “Rice root activity:”, the two dots are not needed in the title. As well as in line 224.

Section four is, more or less, right. Probably some figure with the description of the different mechanisms can help to understand the differences between elements and also between the different oxidation states of the same element.

In general, tables and figures can help to understand the text written.

In my opinion, this is one of the most important conclusions “However, developing rice varieties that can effectively exclude or sequester these metals (As, Cd, and Pb) in multi-metal-contaminated soils requires continuous effort and evaluation”. However, it is necessary to reduce the source of these toxic elements to the rice agrosystems, not only changing the variety of rice.

Finally, minor comments:

When citing like this (line233): [70,71,72,73,28,74] it is better to be consistent with the style of the journal and the order of the references, so this must be [28,70-74]

In the reference I missed many key authors that has been working with these elements, for instance A. Carbonell Barrachina, one of the major specialists in As in the world and worked in paddy systems, among other authors.

Author Response

1. No systematic approach to do the review, for instance PRISMA, was used. No methodology for the review is described in the paper. This is a major issue. 

Answer: Thank you for your valuable feedback. We acknowledge the importance of establishing a systematic review methodology to enhance the rigor and transparency of our analysis. To address this concern, we will incorporate a detailed methodology section outlining the approach used in this review. We appreciate your suggestion to emphasize the relevance of rice agrosystems and their role in global food security. We will revise the introduction to highlight rice as a staple crop and its significance in many countries. Additionally, we will clarify that arsenic (As) is an essential micronutrient for humans, with toxicity dependent on dosage.

2. This review deals with a very important topic when it comes to rice fields, humid agrosystems that, due to their characteristics, can promote the passage of trace elements (toxic elements) to the crop and, consequently, affect the population. So, initially, this is so positive.

However, I would like to make some comments trying to improve the paper for its publication, in the case it will be accepted.

My first suggestion for the introduction is that you can pointed the relevance of the rice agrosystems and the production of rice to feed people, as it is one of the main cereals and basic for many countries. Additionally, it is important to remember that As is an essential element (micronutrient) for humans, the toxicity depends on the dose. (https://www.science.org/content/blog-post/arsenic-rather-lot-when-you-think-about-it) (https://pubmed.ncbi.nlm.nih.gov/18174893/).

Answer: We have revised the introduction section to emphasize the crucial role of rice in food security and nutrition. Additionally, we appreciate the point raised regarding arsenic (As) as a trace element that may have biological functions at low concentrations. We have incorporated this clarification along with relevant references, highlighting that the toxicity of arsenic is dose-dependent. However, it is important to note that arsenic is not an essential nutrient but rather a toxic element. While the human body may require an extremely small amount, the permissible threshold is very low.

3. In the second section, there are several mistakes that should be corrected or clarified.

For instance (line 75): “This semi-etallic element exists in the +3, 0, and +5 oxidation states,” this is wrong, the oxidation states are +5, +3, 0 and -3. See IUPAC table of oxidation states (https://i0.wp.com/www.compoundchem.com/wp-content/uploads/2015/11/The-Periodic-Table-Of-Oxidation-States-2016.png?ssl=1).

Answer: We have corrected this sentence with your comments. 

4. Line 100: Please correct ”Cd2+ has an atomic number of 48, an atomic weight...” It maybe Cd has an...., not the cation.

Answer: We have corrected this sentence properly. 

5. Line 112: the same comment applied to Pb: Pb has an atomic.

Answer: We have corrected this sentence properly. 

6. Section 3 is the most important in the work. In general, the content is so general and probably can be applied to other metals and metalloids. However, the references centered in these three elements are enough good.

Answer: We acknowledge that the discussion in Section 3 is broad and could apply to other metals and metalloids. However, we will ensure that references specifically focus on As, Cd, and Pb to maintain relevance.

7. Line 207: “Rice root activity:”, the two dots are not needed in the title. As well as in line 224.

Answer: We have revised it. 

8. Section four is, more or less, right. Probably some figure with the description of the different mechanisms can help to understand the differences between elements and also between the different oxidation states of the same element.

In general, tables and figures can help to understand the text written.

Answer: We thank the reviewer for the helpful suggestion. We fully agree with your comments and have explicitly outlined the differences between toxic elements, as well as the pathways and mechanisms of uptake, translocation, and accumulation in rice in the referenced figure (Fig. 1).

9. In my opinion, this is one of the most important conclusions “However, developing rice varieties that can effectively exclude or sequester these metals (As, Cd, and Pb) in multi-metal-contaminated soils requires continuous effort and evaluation”. However, it is necessary to reduce the source of these toxic elements to the rice agrosystems, not only changing the variety of rice.

Answer: We appreciate your emphasis on reducing sources of toxic elements in rice agrosystems. We will revise the conclusion to reflect the need for both varietal improvements and the reduction of contamination sources.

10. Finally, minor comments:

When citing like this (line233): [70,71,72,73,28,74] it is better to be consistent with the style of the journal and the order of the references, so this must be [28,70-74]

Answer: We will adjust the citation format to align with the journal’s style, ensuring consistency.

We thank the reviewer again for the valuable feedback and stand ready to address any further comments or suggestions.

Round 2

Reviewer 2 Report

Dear authors, as a review paper, it si necessary to know more than " A comprehensive literature search was
conducted using google scholar, Scopus, PubMed, and Web of Science for articles published between 2000 and 2024". A systematic review includes information about the searching wirds, the inclusion and exclusion criteria and more. That is why I think this review is not consitent.

Several world key authors regarding Cd and As would appear in a systematic review if it would be done. They are not presented in this review. So, my suggestions have been considered only in part.

Author Response

Comment 1: Dear authors, as a review paper, it si necessary to know more than " 
A comprehensive literature search was conducted using google scholar, Scopus, PubMed, and Web of Science for articles published between 2000 and 2024". A systematic review includes information about the searching wirds, the inclusion and exclusion criteria and more. That is why I think this review is not consitent.

Response 1: Thank you for your insightful comment. We acknowledge the distinction between a systematic review and a narrative review. Since our review does not involve data extraction or meta-analysis, it follows a narrative approach, synthesizing and discussing key findings from the literature rather than systematically reviewing them. To enhance transparency, we have clarified this distinction in the manuscript and provided a brief explanation of our literature selection process (see Section 2, Lines 85–96).

Comment 2: Several world key authors regarding Cd and As would appear in a systematic review if it would be done. They are not presented in this review. So, my suggestions have been considered only in part.

Response 2: In revising the manuscript, we have made a concerted effort to include key authors in the field of arsenic and cadmium dynamics in rice systems, such as [Citation No. 50, 101, 109], to ensure that our coverage reflects the broader scientific consensus and significant advancements in the field.

Thank you once again for your time and consideration. We look forward to your feedback.

Round 3

Reviewer 2 Report

Thanks for improving the article following the comments.

I think it is quite goodnow.

No comments